# Frailty Syndrome as a Transition from Compensation to Decompensation: Application to the Biomechanical Regulation of Gait

**DOI:** 10.3390/ijerph20115995

**Published:** 2023-05-30

**Authors:** Lesli Álvarez-Millán, Daniel Castillo-Castillo, Rosa Quispe-Siccha, Argelia Pérez-Pacheco, Maia Angelova, Jesús Rivera-Sánchez, Ruben Fossion

**Affiliations:** 1Doctorado en Ciencias Biomédicas, Universidad Nacional Autónoma de México (UNAM), Mexico City 04510, Mexico; lesli.alvarez26@gmail.com; 2Centro de Ciencias de la Complejidad (C3), Universidad Nacional Autónoma de México (UNAM), Mexico City 04510, Mexico; 3Unidad de Investigación y Desarrollo Tecnológico (UIDT), Hospital General de México Dr. Eduardo Liceaga, Mexico City 06720, Mexico; dcastillocastillo05@gmail.com (D.C.-C.); rosa.quispe@gmail.com (R.Q.-S.); argeliapp@ciencias.unam.mx (A.P.-P.); 4School of Information Technology, Melbourne Burwood Campus, Deakin University, Burwood, VIC 3125, Australia; maia.a@deakin.edu.au; 5Servicio de Geriatría, Hospital General de México Dr. Eduardo Liceaga, Mexico City 06720, Mexico; the_barbarian52@hotmail.com; 6Instituto de Ciencias Nucleares (ICN), Universidad Nacional Autónoma de México (UNAM), Mexico City 04510, Mexico

**Keywords:** frailty, gait, homeostasis, dysregulation, decompensation

## Abstract

Most gait parameters decrease with age and are even more importantly reduced with frailty. However, other gait parameters exhibit different or even opposite trends for aging and frailty, and the underlying reason is unclear. Literature focuses either on aging, or on frailty, and a comprehensive understanding of how biomechanical gait regulation evolves with aging and with frailty seems to be lacking. We monitored gait dynamics in young adults (19–29 years, n = 27, 59% women), middle-aged adults (30–59 years, n = 16, 62% women), and non-frail (>60 years, n = 15, 33% women) and frail older adults (>60 years, n = 31, 71% women) during a 160 m walking test using the triaxial accelerometer of the Zephyr Bioharness 3.0 device (Zephyr Technology, Annapolis, MD, USA). Frailty was evaluated using the Frail Scale (FS) and the Clinical Frailty Scale (CFS). We found that in non-frail older adults, certain gait parameters, such as cadence, were increased, whereas other parameters, such as step length, were decreased, and gait speed is maintained. Conversely, in frail older adults, all gait parameters, including gait speed, were decreased. Our interpretation is that non-frail older adults compensate for a decreased step length with an increased cadence to maintain a functional gait speed, whereas frail older adults decompensate and consequently walk with a characteristic decreased gait speed. We quantified compensation and decompensation on a continuous scale using ratios of the compensated parameter with respect to the corresponding compensating parameter. Compensation and decompensation are general medical concepts that can be applied and quantified for many, if not all, biomechanical and physiological regulatory mechanisms of the human body. This may allow for a new research strategy to quantify both aging and frailty in a systemic and dynamic way.

## 1. Introduction

The frailty syndrome has been theoretically described as a decreased regulatory capacity to maintain homeostasis in the face of stressors [1]. However, there is still considerable uncertainty regarding the concept of frailty and its exact definition, as well as the distinction between frailty, aging, and chronic disease, and the underlying causal pathways, among many other aspects [2]. As discussed in [3] and explained more in detail here below, frailty has been studied using multiple methodologies, each focusing on different facets of the syndrome. The original research approach to frailty is clinical and empirical, where frailty-related signs and symptoms in the patient are quantified using standardized and validated clinical scales [4]. The most popular clinical scales include the Frailty Phenotype (FP), which is based on loss of physical functionality, and is quantified by means of five components: involuntary weight loss, exhaustion, low grip strength, slow gait speed, and low physical activity [5]; the Frailty Index (FI), which calculates the fraction of deficits present in the patient from a list of 70 physical, psychological, and social factors [6]; and other scales such as the Frail Scale (FS) [5], the Clinical Frailty Scale (CFS) [6], and the Edmonton Frail Scale (EFS) [7], which are more applicable in clinical practice because they can be carried out in a short time, are low-cost, and do not require special equipment. A second approach to frailty takes a basic medical science perspective, where genetic, molecular, and cellular biomarkers are analyzed, and deviations of individual biomarkers from their normative ranges are interpreted to reflect physiological dysregulation that eventually results in signs and symptoms of frailty at the clinical level in the older adult [8]. However, neither of these two methodologies explains the core concept of frailty, which are the mechanisms underlying the vulnerability of the organism to stressors [3]. Therefore, inspired by the field of dynamical systems from physics and mathematics, a third approach incorporates the stimulus-response paradigm to evaluate and quantify the resilience of physiological regulation to stressors. This approach was tested theoretically using mathematical models [3] and experimentally by measuring the time to recuperation after acute events in older adults [9].

Physiological regulation appears to be the central concept that connects the extreme sides of the research spectrum on frailty with, on the one hand, the possible genetic, molecular, or cellular origins, and, on the other hand, the clinical consequences of signs and symptoms in the patient. In the present contribution, we focus on the specific variable of gait speed. Gait speed is one of the components of the Frailty Phenotype and is significantly decreased in frail older adults [10,11]. It has been interpreted as a sixth vital sign because a speed of <1 m/s indicates increased dependency in activities of daily living (ADL) [12]. The causal pathway to declining walking performance is not well understood, and possible origins include reduced muscle mass (sarcopenia), decreased muscle contractility (dynapenia), imbalance, and physiological factors, among others [13,14,15]. Gait has the advantage that it can be evaluated non-invasively and continuously in time, using a wide variety of experimental techniques such as video processing and infrared thermography, floor sensors, insole pressure and force sensors, electromyography, and inertial sensors [16,17,18]. Biped gait can be interpreted as a dynamical system [19], with a rhythm that originates from a central pattern generator [20] and is adaptive to perturbations by means of physiological and biomechanical regulation [21]. Gait regulation is quantifiable using time-series analysis [22] and can be modeled mathematically [23]. Analysis of gait dynamics can shed light on the underlying causes of mobility loss as well as suggest clinical interventions for rehabilitation. As explained more in detail here below, much is known in the literature on how gait parameters evolve with aging or frailty, particularly using accelerometry that may be applied in an ambulatory way to reflect activities of daily living (ADL). Average walking speed gradually decreases with age and is reduced below <0.8 m/s with frailty [1,24,25]. It is also well documented, although poorly understood, that step and stride length decrease with aging [26] and with frailty [27]. Cadence is reduced with frailty, but the trend is less clear for aging, where some studies suggest that cadence is maintained [25], whereas other studies even find an increased cadence [28]. The walk ratio or step length vs. step frequency ratio is a speed-independent index of overall neuromotor gait control, reflecting energy expenditure, balance, between-step variability, and attentional demand [29]. The walk ratio is not necessarily related to aging [30], but has been associated with the risk of falling [25] and is not usually applied in frailty research. Acceleration magnitude quantified by the root mean square (RMS) is reduced along the anteroposterior and vertical axes with aging and with frailty [31,32,33]. The RMS is reduced also along the mediolateral axis with frailty [31] but is reduced less [32] or may even be increased with aging [33].

It is clear from the literature that most gait parameters decrease with aging and even more so with frailty. However, it is not clear why other gait parameters show a different or even opposite trend for aging and frailty. Research usually focuses either on aging, comparing young adults, older adults, and/or very old adults [11,25,27,28,34], or on frailty, comparing non-frail, frail, and/or prefrail older adults [11,24,27]. A comprehensive understanding of how biomechanical gait regulation evolves with aging and with frailty seems to be lacking. In this contribution, we report on the results of a gait experiment where multiple study groups, including young and middle-aged control adults and frail and non-frail older adults, walked 160 m at a self-selected speed while being monitored using triaxial accelerometry. This may be the first study where young and older subjects and frail and non-frail subjects are included to study the effects of both aging and frailty. We hypothesized that the opposite trends in gait parameters with aging and frailty may be explained by phenomena of compensation and decompensation, which correspond to abrupt changes in regulatory dynamics, as discussed in multiple areas of medicine. A well-known example from cardiology is compensated heart failure (CHF) after myocardial infarction, where mean systemic blood pressure is increased to compensate for a decreased diastolic filling capacity to ensure a functional cardiac output [35]. Stable chronic heart failure may easily decompensate when specific risk factors place additional strain on the heart muscle and the resulting condition of decompensated heart failure (DHF) implies a worsening of the symptoms including dyspnea, swelling of lower extremities, and fatigue [36]. In the field of gait analysis, compensatory strategies have been observed in gait dynamics to improve stability in the presence of age-related deficits in the physiological function [26,32]. Compensation is also used in the theoretical construct of frailty, although only conceptually and/or qualitatively [37]. The objective of the present contribution was to apply the concepts of compensation and decompensation in a quantitative way to gait dynamics to analyze the evolution of gait parameters with aging and frailty.

## 2. Materials and Methods

### 2.1. Participants

This study was conducted from September 2016 to April 2017 at the Hospital General de México “Dr. Eduardo Liceaga” (HGM). A total of 89 Mexicans participated voluntarily and were divided into four groups, as shown in Table 1: (i) young control adults aged 19–29 years (group C1, n = 27), (ii) middle-aged adults aged 30–59 years (group C2, n = 16), (iii) non-frail older adults over 60 years old (group nF, n = 15), and (iv) frail older adults over 60 years old (group F, n = 31) [38]. In Mexico, people over 60 years old are considered older adults [39]. All study participants signed a letter of informed consent. This protocol was approved by the ethics and research committee of HGM with registry number DI/14/110-B/03/002.

All participating older adults were outpatients of HGM. They were evaluated by an experienced geriatrician using two scales commonly used in clinical practice: the FRAIL scale (FS) [5] and the Clinical Frailty Scale (CFS) [6]. The FS considers older adults to be frail when presenting three or more of the following components: fatigue, absence of resistance (ability to climb one flight of stairs), problems with ambulation (ability to walk one block or 100 m), more than five illnesses, and involuntary weight loss (more than 5%) [5]. The CFS collects information about comorbidities, functionality, and cognitive ability. This allows us to evaluate an older adult on a scale from 1 to 9, where 1 corresponds to a very fit condition and 9 to terminal illness [6]. Older adults were classified as frail when they met the frailty criteria according to both scales; those who did not meet the criteria in either of the scales were classified as non-frail, and older adults who met the criteria in only one of the scales were excluded. The young adults who participated in this study were healthy and functional medical students who did not present any of the risk factors that may be associated with frailty in young adults, such as being homeless [40], drug addiction [41], or having survived childhood cancer [42]. The middle-aged adults who participated in the study were family members and/or caretakers of frail and non-frail older adult participants; they were completely independent and functional, and therefore non-frail. The inclusion and exclusion criteria have been explained in a previous publication [38] and are summarized here. To obtain a more realistic sample of frailty within the older adult population, we included individuals of both genders and with diverse medical histories. The exclusion criteria were Parkinson’s disease, muscular dystrophy, postural vertigo, as well as physical, auditory, visual, or vestibular limitations that put them at risk during the exercise test. For the same reason, people with heart disease, such as severe cardiac failure, severe supraventricular alteration, or monomorphic ventricular extrasystole, were excluded. Exclusion criteria did not include specific alterations such as sarcopenia or cognitive, cardiovascular, or metabolic alterations, instead, we considered such alterations as possible components of the frailty syndrome.

### 2.2. Description of Participant Groups

Here we discuss the characteristics obtained for the population in this study: of the young adult and middle-aged adult control groups, 13 participants (48%) of group C1 and 9 participants (56%) of group C2 performed physical activity (aerobic and anaerobic), on average 1 h 3 times per week. Of the older adults, 13 participants (86%) of group nF performed on average 1 h of daily aerobic exercise, whereas 29 participants (93%) of group F walked on average 800 m daily.

Only participants of group F occasionally used a walking cane (32%). Only two of them used it during the walking test. We included those participants as the results did not change including them or not.

In group C1, 1 participant (4%) smoked regularly and 8 smoked occasionally (30%), whereas in group C2 there were 3 regular smokers (19%). Of the older adults, 10 non-frail participants (67%) and 25 frail participants (80%) were smokers or were exposed to wood smoke.

Only one participant of group C1 had a health condition: asthma. Three participants (18%) of group C2 had hypertension. The health conditions of the older adults were diverse. For group nF, the most common health conditions were systemic arterial hypertension (27%), depression (13%), anxiety (13%), benign prostatic hyperplasia (20%), and atrial fibrillation (13%). Only 2 participants (6%) of group nF did not present any health conditions. The participants of group F had mild cognitive impairment (45%), systemic arterial hypertension (45%), diabetes mellitus type II (32%), depression (29%), anxiety (16%), and chronic heart failure (13%).

### 2.3. Materials

We utilized a Zephyr Bioharness 3.0 device [43] (See Figure 1), which is composed of a band fixed on the chest and various sensors that simultaneously record multiple physiological and biomechanical variables. The raw data included one channel of electrocardiogram (ECG) sampled at 250 Hz, a breathing waveform sampled at 25 Hz, and 3-axial accelerometry sampled at 100 Hz; the derived data included heart rate, breathing rate, breathing amplitude, activity level, and minimum and maximum acceleration in 1 s intervals. In studies on human gait, the accelerometer is often situated as close as possible to the center of mass on the lower back, in which case left and right steps are observed with similar amplitudes [44]. In the case of the Zephyr Bioharness, the accelerometer module is located below the left armpit, resulting in asymmetric accelerometer data. However, this positioning allowed us to differentiate between the left and right steps (See Appendix A). Additionally, we recorded video footage of the gait experiment of each participant to assist in interpreting the accelerometer data. We have partially published the results of the analysis of the physiological variables [38,45]; the present contribution focuses on the analysis of gait regulation using data from 3-axial accelerometry.

### 2.4. Design and Procedure

Anthropometric measures were taken for all participants, including height, weight, and body mass index (BMI) (see Table 1). Additionally, participants were interviewed to collect aspects of their clinical history, physical activity habits, and factors that could affect their pulmonary capacity, such as smoking or frequent exposure to wood smoke. A qualified nurse placed the Zephyr Bioharness 3.0 on each participant, after which participants walked at a self-selected speed along a flat rectangular trajectory on an isolated inner square of HGM. The participants walked the trajectory in one direction and then turned around to complete the trajectory in the other direction, covering a total distance of L = 160 m, see Figure 1 and the description in Ref. [38]. This distance was a choice of convenience, corresponding to the dimensions of the square, but also approximately corresponds to a 3 min walking test, which is a validated test [46]. Older adults walk at approximately 0.89 m/s [47], which corresponds to a distance of 160.2 m (=180 s × 0.89 m/s) for a 3 min walk. The number of steps to cover the whole distance ranged from 200 to 500 steps, depending on step length, and average gait parameters were calculated for the entire trajectory. The study participants did not undergo a pre-conditioning cycle or warming up for two reasons. First, the test is simple and reflects activities of daily living (ADL), such as walking to a local shop, and does not require any specialized practice. Second, the study population includes frail older adults, and any additional activities were avoided that might increase stress or fatigue in this vulnerable population. In most articles where the gait of older adults is studied, a walk is performed over a distance between 0.76 and 6.17 m [48]. However, other articles show the importance of taking longer walking trials as they capture the slow recovery from the perturbations [49].

### 2.5. Calculation of Gait Parameters

We analyzed the acceleration time series and extracted gait parameters along both the time and amplitude axes of each time series. Along the time axis, we calculated the walking duration T of each participant, corresponding to the time interval for which the total acceleration was nonzero. Given the fixed walking distance L = 160 m, the average walking speed corresponded to v = L/T. We calculated the average step duration τ from the first maximum of the autocorrelation function of the vertical accelerometry signal a_VT_ [34] (see Appendix A) and the average cadence, or the number of steps per minute, as c = 1/τ. The total number of steps was N = T/τ, and the average step length was l = L/N [50]. As smaller people tend to walk with smaller steps and higher cadence than taller people, we also calculated normalized cadence and normalized step length, adjusting for the average body height of each population, l_n_ = l. <h>/h and c_n_ = c.√(h/<h>), where h is the height of the individual and <h> is the average height of the corresponding groups C1, C2, nF, or F [51]. We also calculated the walk ratio (WR), or step length-to-step frequency or step length-to-cadence ratio, l/c, and its normalized variant, l_n_/c_n_ [26].

Along the amplitude axis, we studied the magnitude of acceleration in the three directions of movement: anteroposterior a_AP_, mediolateral a_ML_, and vertical a_VT_. For each direction of movement, we calculated the maximum and minimum acceleration per 1 s interval, and our calculated values corresponded well with the peak and min values given by the automatic analysis of the Bioharness 3.0 software. Then, we subtracted these values to estimate the magnitude of movement for every axis, Δa_AP_ = max(a_AP_) − min(a_AP_), and similarly for Δa_ML_ and Δa_VT_. The rationale behind this procedure is as follows: it is known that cadence is approximately 1 stride/s or 2 steps/s [26]. Therefore, Δa_AP_, Δa_ML_, and Δa_VT_ estimate the magnitude of acceleration of individual steps in different directions. We also calculated the root mean square (RMS) of the acceleration signals a_AP_, a_ML_, and a_VT_, which offers an alternative quantification of the magnitude of acceleration per step [35]. However, while Δa_AP_, Δa_ML_, and Δa_VT_ quantify acceleration of individual steps, RMS of the raw accelerometry signal includes whatever movement of the human body. Analogous to the walk ratio l/c, we also calculated acceleration ratios Δa_AP_/Δa_ML_ and Δa_VT_/Δa_ML_.

### 2.6. Statistical Analysis

Numerical variables are described as the mean ± standard error (SE) as they were shown to be normally distributed based on the Shapiro–Wilk test. To compare between groups, we calculated a one-way ANOVA with post hoc Scheffé test for homoscedastic variables (all variables except cadence c and Δa_AP_/Δa_ML_) and Welch’s robustness test with post hoc Games–Howell test for heteroscedastic variables. We used the Statistical Package for the Social Sciences (SPSS) version 22.0 (SPSS Inc., Chicago, IL, USA). A *p*-value ≤ 0.05 was considered significant.

## 3. Results

Table 1 shows results for demographic and anthropometric variables. By construction, age is significantly different between the four study groups. The control groups of young and middle-aged adults are balanced with respect to male and female participants. In the group of non-frail older adults, the majority of participants are males, whereas in the group of frail older adults, the majority are females, which may reflect the fact that frailty has a higher prevalence in women than in men [52]. There are no statistically significant differences between the groups with respect to weight or BMI. Instead, height is significantly decreased in frail older adults (group F) with respect to young adults (group C1), illustrating the importance to consider normalized gait parameters as outlined in Section 2.4.

Table 2, Table 3 and Table 4 and Figure 2, Figure 3 and Figure 4 show results for gait parameters and their evolution with aging and frailty. Step length tends to decrease with aging from group C1 to groups C2 and nF and decreases significantly with frailty (group F). Cadence increases with age from group C1 to C2 to nF, with significant changes for group nF, and then decreases significantly with frailty (group F). The results for non-normalized and normalized step length and cadence are similar. Average walking speed is similar for groups C1, C2, and nF, and is not affected by aging, but walking speed decreases significantly with frailty. A receiver operating characteristic (ROC) analysis shows that a walking speed of 0.83 m/s is the optimal threshold to distinguish between groups nF and F (see Appendix A online). The acceleration magnitude along the three axes demonstrates similar results. Anteroposterior and vertical acceleration magnitudes tend to decrease with age from groups C1 and C2 to groups nF and F, with significant changes for frail older adults (group F). Mediolateral acceleration magnitude tends to increase with age from group C1 to C2 to nF, with significant changes for group nF, and then significantly decreases with frailty (group F). Results for difference (Δ) and root mean square (RMS) measures of acceleration magnitude are similar. Aging does not appear to affect the acceleration vector magnitude, with comparable outcomes for groups C1, C2, and nF, but the acceleration vector magnitude decreases significantly with frailty (group F). Ratios of gait parameters, such as the walk ratio and the acceleration ratios, show a monotonous decreasing trend over groups C1, C2, nF, and F with significant differences for aging and frailty with respect to the control groups of young and middle-aged adults.

## 4. Discussion

The numerical values for the gait parameters obtained in Table 2, Table 3 and Table 4 and Figure 2, Figure 3 and Figure 4 agree with what has been reported in the literature, as will be illustrated in the short review in the following. The average preferred walking speed gradually decreases with age [34] from a typical range of 1.20–1.40 m/s in young adults [28,32,53] to a range of 1.00–1.25 m/s in non-frail older adults [25,26,27,28,30,32] and is reduced to a range of 0.50–0.97 m/s below functional levels with frailty [1,11,24,27,54]. The value of 0.8 m/s has been reported as a threshold to distinguish between frail and non-frail older adults [1]. It is well-documented, although poorly understood, that step length decreases with aging [26,34] from a range of 65–77 cm in young adults [32,53,55] to a range of 56–69 cm in non-frail older adults [11,25,27,28,30] to a range of 49–57 cm with frailty [11,24,27,54]. The evolution of cadence is less clear [34]. Typical values for cadence are 102–117 steps/min in young adults [32,53,55,56]. Cadence appears to be maintained or even increased in non-frail older adults with a range of 103–125 steps/min [25,27,28,28,30,32] and decreases with frailty to a range of 85–109 steps/min [11,24,27,54,57]. The walk ratio tends to decrease with aging from approximately 0.37 in young adults to 0.32 m/s in older adults [30,58] and is not typically used in frailty research. Acceleration magnitude can be quantified by the root mean square (RMS), but magnitudes may not be comparable between sensors placed at different locations on the body. Acceleration magnitudes are significantly reduced with aging along the anteroposterior (AP) and vertical (VT) axes, decreasing from 0.19 (AP) and 0.26 (VT) for young adults to 0.17 (AP) and 0.20 g (VT) for older adults when using an accelerometer placed on the pelvis. The evolution of acceleration magnitude along the mediolateral (ML) axis is less clear and may either decrease from 0.19 for young adults to 0.16 g for older adults [32] or increase from approximately 0.7 to 0.9 g when using an accelerometer placed on the trunk center of mass [33]. Acceleration magnitudes are also reduced with frailty along the anteroposterior (AP) and vertical (VT) axes, decreasing from 0.10 (AP) and 0.13 g (VT) for non-frail older adults to 0.08 (AP) and 0.07 g (VT) for frail older adults when using an accelerometer placed on the lumbar spine. However, acceleration magnitude is not decreased along the mediolateral (ML) axis with frailty and has a typical value of 0.11 g for both frail and non-frail older adults [31].

It is clear from the results of the present contribution, and from the short literature review presented here above, that most gait parameters decrease with aging and even more so with frailty; however, other parameters show a different or even opposite trend for aging and for frailty. In the literature, comparisons are usually made either between groups of non-frail and frail older adults [11,24,27], or between groups of young adults, older adults, and/or very old adults [11,25,27,28,34], and a comprehensive overview of how biomechanical gait regulation evolves with aging and frailty appears to be lacking. As an example, if in Table 2 and Table 3 and Figure 2 and Figure 3, the gait parameters of young adults are taken as normative values, then it may come as a surprise that cadence and mediolateral acceleration are increased above the normal range with aging but return to the expected values for youth and health with frailty. However, the return of individual gait parameters to normative values should not be interpreted as a restoration of the normal health state: biped gait is a regulated process [59], and the interplay between multiple variables is as important as the values of individual parameters. Compensation and decompensation are important concepts in frailty research that take into account such interplay between multiple variables but tend to be discussed only conceptually and qualitatively [37]. We hypothesized that these concepts may help to quantitatively explain the counterintuitive results mentioned before. Taking into account the simple formula that gait speed is the multiplication of step length and cadence, v = l × c, such that a decrease in step length l requires a proportional increase in cadence c to maintain a functional gait speed with v > 1 m/s [60], the significantly increased cadence of the non-frail older adults (group nF) may indeed serve to compensate for the gradual decrease of step length with aging with respect to the young adults (group C1). Frail older adults (group F) have both a significantly decreased step length and a significantly decreased cadence with respect to non-frail older adults (group nF), and although their cadence is similar to that of young adults (group C1), it is not sufficient to maintain a functional gait speed, which may be interpreted as a phenomenon of decompensation. Curiously, the characteristic pattern of “cautious gait” or “senile gait”, with a decreased step length and an increased cadence, appears to be typical for non-frail but not for frail older adults. Clinical studies, such as [30], where gait and aging are analyzed without differentiation according to frailty status, may not be able to discern this typical gait pattern. A similar reasoning appears to apply to acceleration magnitude as well. Mediolateral acceleration has been associated in the literature with step width and area of base support [61]. A gait with a larger base support may serve as a strategy to gain stability in the face of perturbations, e.g., neurocognitive alterations, reduced energy, decreased mass or strength, or the presence of obstacles [62]. We interpret the significantly increased mediolateral acceleration Δa_ML_ of the non-frail older adults (group nF) as a reflection of an increased base support to compensate for reduced stability. Frail older adults (group F) have significantly decreased mediolateral acceleration compared to non-frail older adults (group nF). Although the mediolateral acceleration of frail older adults (group F) is similar to that of young adults (group C1), their base support may not be sufficient to ensure adequate stability and may explain the increased risk of falls in frail older adults [63], which we interpret as a phenomenon of decompensation. The effect of compensation by lateral movement is known as a “wide-based gait” or “waddle” and occurs in the face of both internal perturbations, such as in the case of pregnant women [64] or obesity [65], or external perturbations, such as ship passengers [66] or train conductors [62]. Again, a wide-based gait appears to be more characteristic of non-frail older adults (group nF) than frail older adults (group F).

In contrast to the opposite trends of the gait parameters c and Δa_ML_ with aging and frailty discussed so far, the ratios of gait parameters l/c, Δa_AP_/Δa_ML_, and Δa_VT_/Δa_ML_ of Table 4 and Figure 4 show a monotonous decreasing trend over all four populations from no compensation (groups C1 and C2) to compensation (group nF) and finally to decompensation (group F). The walk ratio l/c has been interpreted as a measure of the quality of the overall neuromotor gait regulation [29], and the same interpretation may be valid as well for Δa_AP_/Δa_ML_ and Δa_VT_/Δa_ML_ given their very similar behavior. Here, we propose the ratios l/c, Δa_AP_/Δa_ML_, and Δa_VT_/Δa_ML_ as concrete and practical metrics to quantify compensation and decompensation in gait dynamics. The rationale is the following: each of these ratios evaluates a gait parameter that is being compensated with respect to the gait parameter that is actively performing the compensation; therefore, both an increased compensation and an underperforming regulatory mechanism (decompensation) result in a reduced ratio. We speculate that the underlying reason for a transition from compensation with aging to decompensation with frailty is due to homeostatic effort, i.e., the extra energy needed for compensation and increased load to keep a specific regulatory mechanism working [67]. These additional energy requirements are taken from the physiological reserves, such that there must be a proportionality relation between how much a system compensates and how much of the physiological reserves are in use. Such a reduction in available physiological reserves has been called presbyhomeostenosis [68,69]. When reserves are no longer available, the only possible outcome is decompensation, describing an energetic pathway to mobility loss and slowing down of gait speed [67].

Several authors have emphasized multiple conceptual similarities in physiological and biomechanical regulation for a wide variety of body mechanisms. Not only are most regulatory mechanisms based on negative feedback, but also the variables that participate in these mechanisms can be subdivided into two broad categories: there are the so-called “regulated variables” that represent the stable internal environment or “milieu intérieur” of Claude Bernard, such as blood volume, blood flow, core temperature, blood glucose concentration, blood oxygen saturation, etc., and each of these regulated variables has a variety of associated “effector variables” that are responsible for adapting to perturbations from the internal and external environments with an objective to maintain the stability of their regulated variable [70,71,72,73]. The example of compensated (CHF) and decompensated heart failure (DHF) that was mentioned in the Introduction section illustrates that the role of effector variables (in this case: heart diastolic capacity and systemic blood pressure) is not only to respond to internal or external perturbations but also to mutually compensate to ensure the stability of the associated regulated variable (here: cardiac output). Recently, it has become clear that the similarities between different physiological and biomechanical regulatory mechanisms are more than just conceptual but can be quantified as well. Time series analysis shows that in optimal conditions of youth and health, regulated variables and effector variables have specific statistical properties that deviate in predictable ways with adverse conditions of aging and/or disease [74,75,76,77,78,79]. In the present context of biomechanics of gait regulation, step length and cadence can be interpreted as two effector variables that are responsible for the functionality of the associated regulated variable of gait speed, while mediolateral acceleration, step width, and base of support are effector variables that play a role in maintaining the regulated variable of equilibrium. Whereas in clinical practice, regulatory mechanisms of different body systems tend to be evaluated with specialized tools that may not be applicable to other systems, evaluating the effects of compensation and decompensation is promising because of the general applicability. Similar to the present application in gait dynamics, compensation, and decompensation may be quantified in other regulatory mechanisms by calculating the ratio of the compensated effector variable with respect to the compensating effector variable. Ratios appear to be powerful computational tools to assess the performance of regulatory mechanisms in physiology, as has been shown before, e.g., in cardiology [74,80]. One of the most important challenges to evaluate aging and frailty is that multiple body systems are affected, and often alterations only become visible when they are stressed. We propose to study aging and frailty by assessing compensation and decompensation across the most important regulatory systems of the body, including the cardiovascular, respiratory, metabolic, thermoregulatory, and locomotor systems. This approach would have the advantage of allowing for both a systemic and a dynamic evaluation of the human body.

### Strengths, Limitations, and Implications of This Study

Strengths: The present contribution may be the first study to include groups of young adults, middle-aged adults, non-frail, and frail older adults to examine the effects of both aging and frailty on gait dynamics. The careful application of selection criteria and assessment procedures has enhanced the internal validity of our study. Whereas compensation and decompensation are only discussed conceptually and qualitatively in the literature, here we have proposed a quantitative metric based on the ratio between a compensated variable and the associated compensating variable, that can be universally applicable to different physiological or biomechanical regulatory mechanisms.

Limitations: The sample size of our study was relatively small, with only 89 participants. Given that populations of non-frail and frail older adults are rather heterogeneous, a larger population size might be needed to improve the external validity. The young adult and frail older adult groups were approximately twice as large as the middle-aged adult and non-frail older adult groups, such that the groups are rather unbalanced. In this exploratory study, we decided not to include a pre-frail group since various gait parameters, in particular gait speed, are not altered with respect to non-frail older adults, and we focused our attention on the comparison between frail and non-frail older adults. It may be interesting to study the effects of compensation and decompensation in pre-frail older adults in future studies.

Implications: Frail older adults seem to have lost the ability to compensate adequately for suboptimal gait parameters, resulting in a reduction of gait speed. We propose examining the compensatory capacity of other biomechanical and physiological regulatory mechanisms to determine whether a transition from compensation to decompensation is a common characteristic of frailty. If so, this could provide an alternative approach to studying frailty in a systemic and dynamic manner.

## 5. Conclusions

Most gait parameters decrease with aging, and even more so with frailty. It is not clear why other gait parameters show different or even opposite trends for aging and frailty. The scientific literature tends to focus either on aging or frailty, and a comprehensive understanding of how biomechanical gait regulation evolves with aging and frailty seems to be lacking. This contribution may be the first study where young adults, middle-aged adults, and non-frail and frail older adults were included to study the effects of both aging and frailty. Non-frail older adults appear to increase specific gait parameters, such as cadence, to compensate for other underperforming gait parameters, such as step length, to maintain functional gait speed. Biomechanical regulation in frail older adults appears to decompensate because all gait parameters are decreased, and walking speed drops below the threshold of functionality. Compensation and decompensation are general medical concepts that can be applied and quantified for many, if not all, of the biomechanical and physiological regulatory mechanisms of the human body, and may allow for a systemic and dynamic evaluation of aging and frailty.

## Figures and Tables

**Figure 1 ijerph-20-05995-f001:**
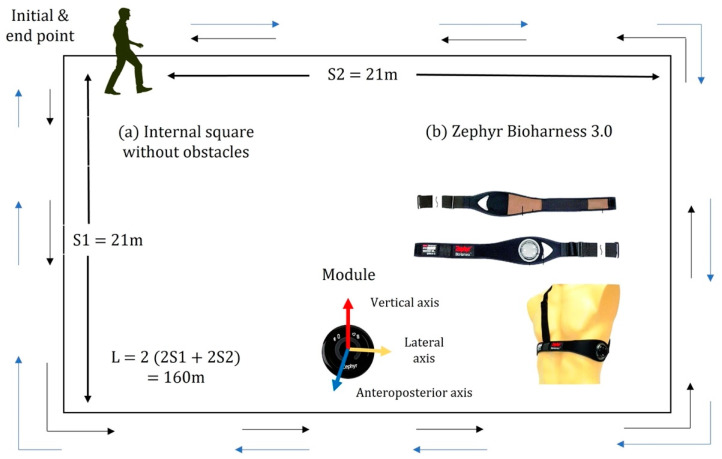
(**a**) Diagram of the experimental area with sides S1 = 19 and S2 = 21 m and total distance L = 2 (2S1 + 2S2) = 160 m, (**b**) the physiological and biomechanical monitoring device Zephyr BioHarness 3.0 consisting of a chest strap and a bioModule containing the internal memory and battery. The anteroposterior (AP), medio-lateral (ML), and vertical (VT) axes of triaxial accelerometry are also indicated. Modified from Zephyr Technology Corporation, Annapolis, MD, USA—a division of Medtronic.

**Figure 2 ijerph-20-05995-f002:**
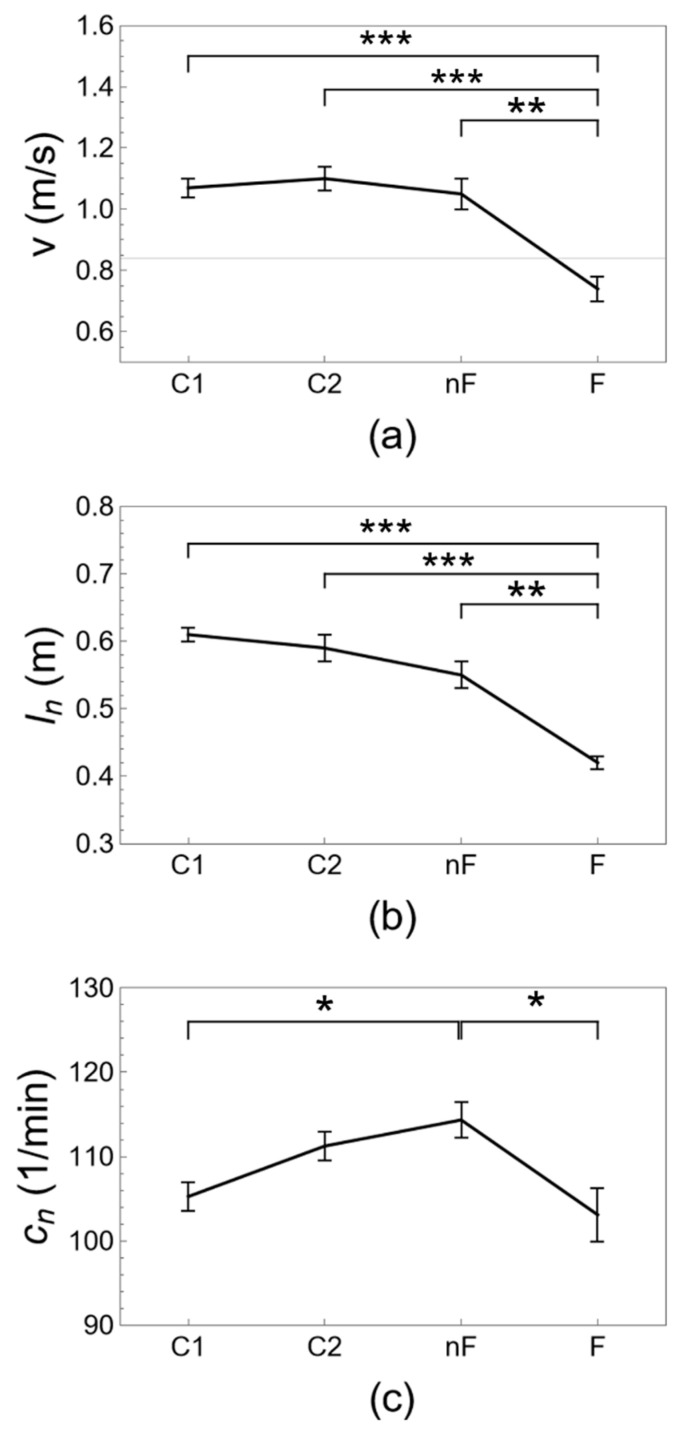
Gait parameters along the time axis, (**a**) velocity v, (**b**) normalized step length l_n_ and (**c**) normalized cadence c_n_. The evolution of gait parameters is shown for young adults (group C1), middle-aged adults (group C2), non-frail older adults (group nF), and frail older adults (group F). Indicated are mean ± standard error, pairwise statistically significant differences with *p* < 0.05 (*), *p* < 0.005 (**), *p* < 0.001 (***), and the gait velocity threshold (v = 0.83 m/s) for frailty obtained with ROC analysis (horizontal gridline).

**Figure 3 ijerph-20-05995-f003:**
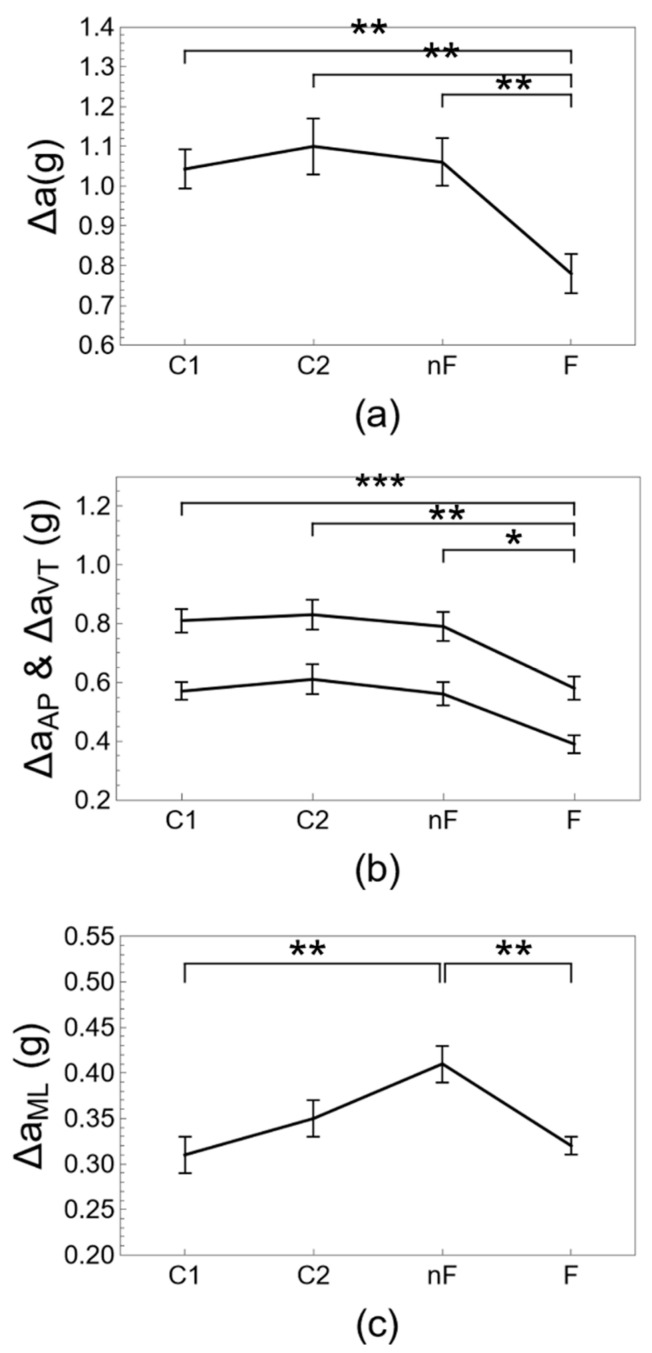
Gait parameters along the magnitude axis, (**a**) vector acceleration Δa, (**b**) anteroposterior acceleration Δa_AP_, and vertical acceleration Δa_VT_, and (**c**) mediolateral acceleration Δa_ML_. The evolution of gait parameters is shown for young adults (group C1), middle-aged adults (group C2), non-frail older adults (group nF), and frail older adults (group F). Indicated are mean ± standard error, and pairwise statistically significant differences with *p* < 0.05 (*), *p* < 0.005 (**), *p* < 0.001 (***).

**Figure 4 ijerph-20-05995-f004:**
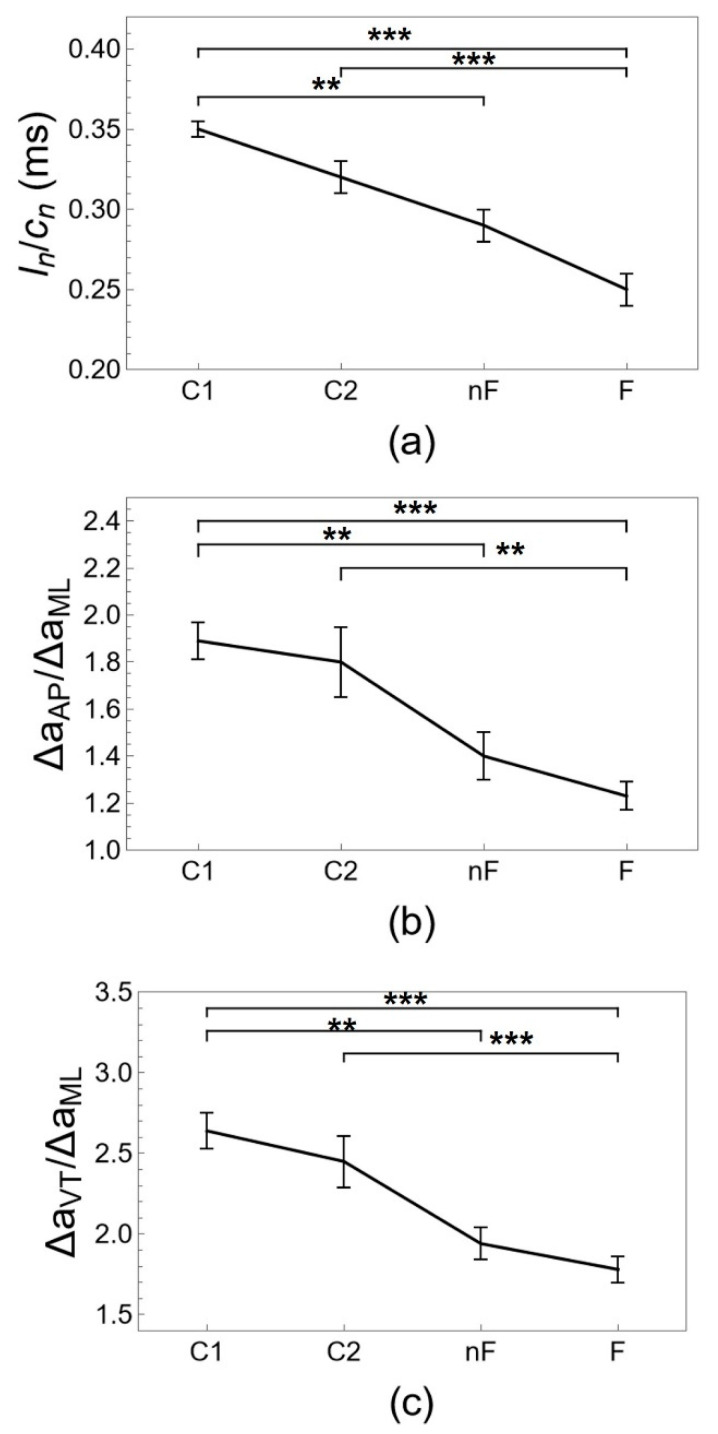
Ratios of gait parameters, (**a**) normalized walk ratio l_n_/c_n_, (**b**) acceleration ratio Δa_AP_/Δa_ML_, and (**c**) acceleration ratio Δa_VT_/Δa_ML_. The evolution of gait parameters is shown for young adults (group C1), middle-aged adults (group C2), non-frail older adults (group nF), and frail older adults (group F). Indicated are mean ± standard error, and pairwise statistically significant differences with *p* < 0.05 (*), *p* < 0.005 (**), *p* < 0.001 (***).

**Table 1 ijerph-20-05995-t001:** Demographic and anthropometric measures (mean and standard error) are shown for each group. The sample size is represented with n. Results are given for young control adults (group C1), middle-aged control adults (group C2), non-frail older adults (group nF), and frail older adults (group F).

Variable	Group C1 (n = 27)	Group C2 (n = 16)	Group nF (n = 15)	Group F (n = 31)	*p*-Value between All Groups
Age (years) (% female)	22.3 ± 0.4 (59%)	48.5 ± 2.2 (62%)	72.7 ± 2.5 ^¶,#^ (33%)	78.5 ± 1 ^¶,#^ (71%)	0.000
Weight (kg)	66.8 ± 2.4	70.7 ± 4.4	69.3 ± 3.1	61.6 ± 2.2	0.154
Height (m)	1.7 ± 0.02	1.6 ± 0.02	1.6 ± 0.02	1.5 ± 0.02 ^¶^	0.000
BMI (kg/m^2^)	23.9 ± 0.6	27.4 ± 1.5	26.8 ± 1.03	26.3 ± 0.7	0.060

^¶^ *p* < 0.05 compared to C1; ^#^ *p* < 0.05 compared to C2; ^§^ *p* < 0.05 compared to nF.

**Table 2 ijerph-20-05995-t002:** Gait parameters along the time axis. Shown are gait speed (v), average step length (l), normalized average step length ($l_{n}$), average cadence (c), and normalized average cadence ($c_{n}$). Results are given for young control adults (group C1), middle-aged control adults (group C2), non-frail older adults (group nF), and frail older adults (group F).

Variable	Group C1 (n = 27)	Group C2 (n = 16)	Group nF (n = 15)	Group F (n = 31)	*p*-Value between All Groups
v (m/s)	1.08 ± 0.03	1.09 ± 0.05	1.05 ± 0.06	0.74 ± 0.05 ^¶,#,§^	0.000
l (m)	0.61 ± 0.01	0.59 ± 0.04	0.55 ± 0.02	0.42 ± 0.03 ^¶,#,§^	0.000
l_n_ (m)	0.61 ± 0.01	0.6 ± 0.02	0.55 ± 0.02	0.42 ± 0.02 ^¶,#,§^	0.000
c (1/min)	105.26 ± 1.72	111.25 ± 1.78	114.35 ± 3.21 ^¶^	103.1 ± 3.1 ^§^	0.005
c_n_ (1/min)	105.64 ± 1.7	110.62 ± 1.7	114.05 ± 2.1 ^¶^	103.53 ± 3.2 ^§^	0.005

^¶^ *p* < 0.05 compared to C1; ^#^ *p* < 0.05 compared to C2; ^§^ *p* < 0.05 compared to nF.

**Table 3 ijerph-20-05995-t003:** Gait parameters along the amplitude axis. Shown are amplitudes of the acceleration vector (Δa), anteroposterior acceleration (Δ$a_{AP}$), vertical acceleration (Δ$a_{VT}$) and mediolateral acceleration (Δ$a_{ML}$), root mean square of vector acceleration RMS(a), anteroposterior acceleration RMS($a_{AP})$, vertical acceleration RMS($a_{VT}$), and mediolateral acceleration RMS($a_{ML})$. Results are given for young control adults (group C1), middle-aged control adults (group C2), non-frail older adults (group nF), and frail older adults (group F).

Variable	Group C1 (n = 27)	Group C2 (n = 16)	Group nF (n = 15)	Group F (n = 31)	*p*-Value between All Groups
Δa (g)	1.04 ± 0.05	1.10 ± 0.07	1.06 ± 0.06	0.80 ± 0.05 ^¶,#,§^	0.000
Δa_AP_ (g)	0.57 ± 0.03	0.61 ± 0.05	0.56 ± 0.04	0.39 ± 0.03 ^¶,#,§^	0.000
Δa_VT_ (g)	0.81 ± 0.04	0.83 ± 0.05	0.79 ± 0.05	0.58 ± 0.04 ^¶,#,§^	0.000
Δa_ML_ (g)	0.31 ± 0.02	0.35 ± 0.02	0.41 ± 0.02 ^¶^	0.32 ± 0.01 ^§^	0.002
RMS(a) (g)	0.12 ± 0.007	0.13 ± 0.008	0.12 ± 0.008	0.09 ± 0.006 ^¶,#,§^	0.000
RMS($a_{AP}$) (g)	0.13 ± 0.007	0.14 ± 0.01	0.14 ± 0.008	0.1 ± 0.006 ^¶,#,§^	0.001
RMS($a_{VT}$) (g)	0.2 ± 0.01	0.22 ± 0.02	0.22 ± 0.02	0.14 ± 0.01 ^¶,#,§^	0.000
RMS ($a_{ML}$) (g)	0.08 ± 0.003	0.09 ± 0.004	0.1 ± 0.003 ^¶^	0.09 ± 0.003	0.002

^¶^ *p* < 0.05 compared to C1; ^#^ *p* < 0.05 compared to C2; ^§^ *p* < 0.05 compared to nF.

**Table 4 ijerph-20-05995-t004:** Ratios of gait parameters. Shown are the walk ratio l/c, the normalized walk ratio $l_{n}$/$c_{n}$, and the acceleration ratios Δ$a_{AP}$ /Δ$a_{ML}$ and Δ$a_{VT}$ /Δ$a_{ML}$. Results are given for young control adults (group C1), middle-aged control adults (group C2), non-frail older adults (group nF), and frail older adults (group F).

Variable.	Group C1 (n = 27)	Group C2 (n = 16)	Group nF (n = 15)	Group F (n = 31)	*p*-Value between All Groups
l/c (m.s)	0.35 ± 0.006	0.32 ± 0.012	0.29 ± 0.01 ^¶^	0.25 ± 0.01 ^¶,#^	0.000
l_n_/c_n_ (m.s)	0.34 ± 0.006	0.32 ± 0.01	0.29 ± 0.01 ^¶^	0.25 ± 0.01 ^¶,#^	0.000
Δa_AP_/Δa_ML_	1.9 ± 0.08	1.8 ± 0.15	1.4 ± 0.1 ^¶^	1.2 ± 0.06 ^¶,#^	0.000
Δa_VT_/Δa_ML_	2.6 ± 0.11	2.4 ± 0.16	1.9 ± 0.1 ^¶^	1.8 ± 0.08 ^¶,#^	0.000

^¶^ *p* < 0.05 compared to C1; ^#^ *p* < 0.05 compared to C2; ^§^ *p* < 0.05 compared to nF.

## Data Availability

The data presented in this study are available on request from the corresponding author.

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
