# Peer review of "Frailty Syndrome as a Transition from Compensation to Decompensation: Application to the Biomechanical Regulation of Gait"

_ijerph, 2023, doi:10.3390/ijerph20115995_

Round 1
Reviewer 1 Report (Previous Reviewer 1)
The paper can be accepted in the present form. the authors have addressed all of this reviewer's comments in this resubmitted draft.
Author Response
Manuscript ID: ijerph-2095577 “Frailty syndrome as a transition from compensation to decompensation: Application to the biomechanical regulation of gait”
Response to Reviewer 1
Comment: The paper can be accepted in the present form. the authors have addressed all of this reviewer's comments in this resubmitted draft.
Response: Thank you for your comments, which helped us to improve our manuscript.

Reviewer 2 Report (New Reviewer)

Author Response
Manuscript ID: ijerph-2095577 “Frailty syndrome as a transition from compensation to decompensation: Application to the biomechanical regulation of gait”
Response to Reviewer 2
First, I would like to congratulate the excellent work that was presented. I understand how
laborious it is to prepare a manuscript for publication and we must give proper credit.
Congratulations.
You've done a good job. However, I have a few questions to ask.
The following comments refer to the introduction of the manuscript.
Comment: Line 16 – What kind of regulation? Cardiovascular regulation? Please, provide enough information in this sentence.
Comment: Lines 16-17 – Dysregulation of what? You need to provide clear information.
Comment: Line 19 – Multiple variables – which variables? Also, “specific regulatory mechanism”. Which mechanism?
Comment: Lines 34-37 – This sentence is too long. Please, try to write in 2 or 3 sentences.
Comment: Lines 38 – 39 – This sentence is out of context. Please, provide more information and add some references.
Comment: Lines 39 – 41 – What do you call “generic effects”?
Comment: Lines 49 – Please, instead of using “having each their own understanding of frailty”, bring the main difference between each concept.
Comment: Are you sure that the Fried Phenotype has “particular reduced gait speed”? How about the other 4 criteria?
Comment: Lines 52 – 54 – Frailty Index is one of the most used tools to classify frail people in clinical practice. You can organize the set of questions according to your reality. I suggest changing this sentence.
Comment: To apply such questionnaires, every evaluator undergoes training. Therefore, the items considered subjective can be observed and discussed as a way of improvement. As such, they are as highly effective as other tools for identifying people with frailty. I suggest changing this sentence.
Comment: Lines 58 – 61 – This sentence is too long. Please, rewrite.
Comment: Line 61- What kind of biomarkers?
Comment: Lines 61 – 69 - There are studies that analyze blood biomarkers in people with frailty during stressors (e.g., exercise) that provide evidence of physiological (e.g., immune) system decline in frail people. I suggest rewriting the sentence.
Comment: Please be more precise with this information: "...we are interested in generic aspects of Please be more precise with this information: "...we are interested in generic aspects of regulatory mechanisms".regulatory mechanisms".
Comment: Lines 74 – 75 - Please be more precise with this information:
"recovery times of these mechanisms": which mechanisms?
"interplay between multiple variables" which variables?
"within a specific mechanism" which mechanisms?
Comment: Lines 73 – 93 – This sentence should be rewritten. Some information is too vague. Please, provide information that has connections and be clear.
Comment: Line 94 - It's the first time you mention the biomechanics of gait regulation. This information must appear at the beginning of the introduction. The text started to become clearer now.
Comment: I strongly suggest rewriting the entire introduction. There is a lack of information and at times there is no clarity in what is being written. I suggest focusing on the decline in gait speed in people with frailty and how it is possible to use accelerometers to assess this variable. Also, I suggest not writing "the research question" "the research hypothesis". Be more precise with the information.
General response for the introduction section: Thank you for your comments, which helped us to improve our manuscript. We have revised the introduction as you suggested and write a new version of this section. In this updated version, less is mentioned about general concepts, and more focus is given to the concept of gait regulation. See lines 30-141.
Comment: Line 132 – What is “mature adults”?
Response: We have changed the term "mature adults" to "middle-aged adults" who are participants between the ages of 30 and 59 (see line 147).
Comment: Lines 136 – 145 - Please provide a solid reason why the youth group did not take the same test as the older adults (Clinical Frailty Scale). Although it is difficult, there are young people who can be considered pre-frail or frail. This condition is not based on chronological age, but on biological age. Please I would like to read a solid reason for this.
Response: Chronological age is the most important independent risk factor for developing frailty, but, as the referee argues, it is not the only one. Risk factors that can cause frailty in young adults include being homeless [36], drug addiction [37], and survivors of childhood cancer [38]. The young adults who participated in this study were healthy and functional medical students who did not have any of the aforementioned risk factors. The middle-aged adults who participated in the study were family members and/or caretakers of both frail and non-frail older adult participants. They were completely independent and functional and therefore, non-frail.
We included this information on line 164-172.
Comment: Line 137 – Please, give a brief explanation what which scale measures.
Response: The manuscript mentions five frailty scales, three of which are only cited as the most commonly used clinical scales. The other two were used to assess frailty in the elderly in this study:
- The frailty phenotype (FP) on lines 39-41 is based on loss of physical functionality and is quantified by means of five components: involuntary weight loss, exhaustion, low grip strength, slow gait speed, and low physical activity [3].
- The frailty index (FI) on lines 42-43 takes into consideration the accumulation of age-associated deficits [4].
- The Edmonton Frail Scale (EFS) on line 44 considers aspects of cognition, functional independence, social support, medication use, nutrition, mood, continence, and functional performance [6].
The two scales used to evaluate frailty in this study were:
- The FRAIL scale (FS) on lines 157-159 identifies older adults as frail if they exhibit three or more of the following components: fatigue, absence of resistance (ability to climb one flight of stairs), problems with ambulation (ability to walk one block or 100m), more than five illnesses, and involuntary weight loss (more than 5%).
- The Clinical Frailty Scale (CFS) on lines 159-161 collects information about comorbidities, functionality, and cognitive ability. This places the patient on a scale from 1 to 9, where 1 corresponds to a very fit subject and 9 to one with a terminal illness.
Comment: Line 142 – only “frail”. How about the pre-frail condition?
Response: In this exploratory study, we decided not to include the pre-frail population since a number of specific gait parameters, in particular gait speed, are not altered in this group with respect to the control populations (young adults and middle-aged adults). Therefore, we analyzed the two groups where the differences are most notable (frail vs non-frail).
We included this on lines 297-300.
Comment: Line 162 – 163 - It is not clear: is young considered C1 and mature control groups C2?
Response: Yes. We have already changed mature control for middle aged group but both groups are defined on lines 146-147.
Comment: Lines 162 – 179 – How did you get all those information? You need to provide clear information about that.
Response: These were the characteristics observed in the population of this study. We have clarified that in the beginning of that section on line 303.
Comment: Line 321 – I suggest starting the sentence with the main findings.
Response: We have rewritten the discussion to provide a clearer interpretation of the findings. See lines 517-697.
Comment: Lines 406 – 415 – I suggest rewriting the conclusion. Most of the information has been provided before.
Response: We have rewritten the conclusion summarizing the main findings. See lines 858-871.

Reviewer 3 Report (New Reviewer)
The author submitted the paper " Frailty syndrome as a transition from compensation to decompensation: Application to the biomechanical regulation of gait" with innovative research and reference value for practical applications. However, some considerations that should be notice.
Abstract:
Please specific the purpose of this research.
Introduction
Lines 38-40: Is there any literature support for the above?
Lines 81-84: Is there any literature support for the above?
Lines 85-88: Is there any literature support for the above?
Lines 94-99: Is there any literature support for the above?
Lines 104-106: Is there any literature support for the above?
Lines 112-123: A specific the purpose of the study is need.
No hypothesized in introduction?
Methods
Line 126-131: Usually the term "years old" will not be use as a proper noun. the "yo" is unnecessary.
Line 186: Please indicate the detail information of device "Zephyr Bioharness" and the sampling rate of those data?
Line 206: check the “)” seems unnecessary.
Line 201: Please show the distance of all sides in Figure 1.
Line 215: Did any prepare before they walk such as warm up or stretch?
Line 217-219: why 200-500 steps is there any reason or literature support?
Line 242: what is normalized cadence? Please explain.
Results
Results were showed in three parts (Table 2). However, it's seems confusion. Suggest use separate table present the results and connect with the depiction.
Simplify the figures, show the major findings
Discussion/Conclusions
Line 321-324: Is there any literature support for the above?
Line 327:"...return to the normal values..." how to define the "normal values"?
Line 337-339: “the purposes of the biomechanical regulation of gait is to maintain a minimum functional gait speed of v > 1m/s to be able to take part in the independent activities of daily living.” Is there any literature indicted the relation of gait speed and independent activities?
Line 340-343: The hypothesized shouldn’t in discussion.
Line 356-358: What energy and physiological reserves were evaluated that support the sentence.
Line 376-378: Same question, how did you explain the energy issue? Any tests or literature?
Line 396-399: Please replenish the information of experiment settings and results in manuscript.
What limitations does the study present? And what implications for practice has the study developed? What suggestions for future studies are there, in addition to those presented?
Author Response
Manuscript ID: ijerph-2095577 “Frailty syndrome as a transition from compensation to decompensation: Application to the biomechanical regulation of gait”
Response to Reviewer 3
The author submitted the paper " Frailty syndrome as a transition from compensation to decompensation: Application to the biomechanical regulation of gait" with innovative research and reference value for practical applications. However, some considerations that should be notice.
Abstract
Comment: Please specific the purpose of this research.
Response: Thank you for your comments, which helped us to improve our manuscript. At the suggestion of reviewer 2, we rewrite some sections of the manuscript, taking care not to alter the content, but rather to improve its understanding. The new version of the abstract can be found on the lines 16-27.
Introduction
Lines 38-40: Is there any literature support for the above?
Lines 81-84: Is there any literature support for the above?
Lines 85-88: Is there any literature support for the above?
Lines 94-99: Is there any literature support for the above?
Lines 104-106: Is there any literature support for the above?
Lines 112-123: A specific the purpose of the study is need.
No hypothesized in introduction?
General response for the introduction section: We have revised the introduction as suggested by referee 2 and write a new version of this section. In this updated version, literature support is given. See lines 30-141. And the hypothesis can be found on lines 136-139.
Methods
Comment: Line 126-131: Usually the term "years old" will not be use as a proper noun. the "yo" is unnecessary.
Response: We have corrected that on lines 146-147.
- i) young control adults aged 19-29 years (C1, n=27), ii) middle-aged adults from 30-59 years (C2, n=16)
Comment: Line 186: Please indicate the detail information of device "Zephyr Bioharness" and the sampling rate of those data?
Response: The Zephyr Bioharness is a telemetric device that simultaneously measures several physiological signals. Raw data includes one channel of electrocardiogram (ECG) with a sample frequency (fs) of 250 Hz, the breathing waveform with fs = 25 Hz, and 3-axial accelerometry with fs = 100 Hz. We have included the sample frequency of accelerometry on lines 335-336 since it is the variable analyzed in this manuscript.
Comment: Line 206: check the “)” seems unnecessary.
Response: We have eliminated the “)”.
Comment: Line 201: Please show the distance of all sides in Figure 1.
Response: We included the distances in the figure which is located on line 348.
Comment: Line 215: Did any prepare before they walk such as warm up or stretch?
Response: The participants did not undergo a preconditioning cycle for two reasons. First, the study population includes frail older adults, and any additional activities that might increase stress or fatigue in this vulnerable population were avoided. Secondly, they did not warm up or stretch because the test is straightforward and reflects activities from daily life, such as walking one or more blocks to a local shop, and does not require any specialized preparation.
We included that information on lines 370-374.
Comment: Line 217-219: why 200-500 steps is there any reason or literature support?
Response: The number of steps is a consequence of choosing a 160m distance, which corresponds to walking twice the perimeter of the internal area used to carry out the experiment [34]. The distance of 160m was selected based on the 3-minute walking test, which is a validated test [41]. Older adults walk at approximately 0.89 m/s [42], which corresponds to a distance of 160.2 m (= 180 s × 0.89 m/s) for a 3-minute walk.
See lines 366-370.
Comment: Line 242: what is normalized cadence? Please explain.
Response: To account for the effect of body height on walking, we calculated normalized cadence and normalized step length in our study. This was necessary because shorter individuals typically take smaller steps and walk with a higher cadence than taller individuals. To adjust for these differences, we normalized the cadence and step length measurements by dividing them by the average body height of each population. This allowed us to compare the walking patterns of individuals across different heights on an equal footing. This is mentioned on lines 392-394.
Results
Comment: Results were showed in three parts (Table 2). However, it's seems confusion. Suggest use separate table present the results and connect with the depiction.
Response: We have separated the original table 2 in 3 new tables with the information of gait parameters calculated from the time axis (Table 2), the gait parameters calculated from the amplitude axis (Table 3) and finally, the ratios of gait parameters (Table 4). They are included in the manuscript on lines 450-469.
Comment: Simplify the figures, show the major findings
Response: We separated the original Figure 2, in two figures, where the new Figure 2 on lines 469-497 includes the velocity of gait and the related variables step length and cadence. The new Figure 3 includes the accelerometry parameters of gait in the three axis of movement. It is on lines 503-507.
Discussion/Conclusions
Comment: Line 321-324: Is there any literature support for the above?
Response: We have included the specific values reported in the literature in comparison to our results. See lines 518-530.
Comment: Line 327:"...return to the normal values..." how to define the "normal values"?
Response: We have changed the phrase “normal values”, and instead, make a comparison of our results with the values reported in the literature. See lines 518-51430 are reported the ranges of values found in various scientific studies to make a comparison.
Comment: Line 337-339: “the purposes of the biomechanical regulation of gait is to maintain a minimum functional gait speed of v > 1m/s to be able to take part in the independent activities of daily living.” Is there any literature indicted the relation of gait speed and independent activities?
Response: We have included the reference on line 588.
Fielding, R. A., Vellas, B., Evans, W. J., Bhasin, S., Morley, J. E., Newman, A. B., ... & Zamboni, M. (2011). Sarcopenia: an undiagnosed condition in older adults. Current consensus definition: prevalence, etiology, and consequences. International working group on sarcopenia. Journal of the American Medical Directors Association, 12(4), 249-256.
Comment: Line 340-343: The hypothesized shouldn’t in discussion.
Response: We included the hypothesis in the introduction, now, we simply take it up again in this section to discuss it with respect to the results found.
Comment: Line 356-358: What energy and physiological reserves were evaluated that support the sentence. Line 376-378: Same question, how did you explain the energy issue? Any tests or literature?
Response: we did not measure energy or physiological reserves in this study, so they are not presented as quantitative results. In the discussion section they are mentioned, since it is a well-known effect in the literature. We cited one of the most popular articles on line 626.
Schrack, J. A., Simonsick, E. M., & Ferrucci, L. (2010). The energetic pathway to mobility loss: an emerging new framework for longitudinal studies on aging. Journal of the American Geriatrics Society, 58, S329-S336.
Comment: Line 396-399: Please replenish the information of experiment settings and results in manuscript.
Response: We have restated the discussion and focused on better explaining the results obtained from gait, as well as enhancing the understanding of the concepts of compensation and decompensation. Therefore, we decided to eliminate the comment on lines 396-399 of the previous version of the manuscript, which may not be as relevant at this time and could cause confusion.
Comment: What limitations does the study present? And what implications for practice has the study developed? What suggestions for future studies are there, in addition to those presented?
Response:
Strengths: The present contribution may be the first study to include groups of young adults, middle-aged adults, non-frail, and frail older adults to examine the effects of both aging and frailty on gait dynamics. We proposed a metric to quantify the degree of compensation and decompensation, which are crucial concepts to understand frail-ty, but which are only discussed in a qualitative way in the literature.
Limitations: The sample size was relatively small and unbalanced, with only 89 partic-ipants. And with the young adult and frail older adult groups being approximately twice as large as the middle-aged adult and non-frail older adult groups. This study did not include pre-frail older adults. For future studies, it is recommended to expand the sample and include this population. In this study, we included older adults with different conditions. However, it would be interesting to study a larger population to include a more heterogeneous sample. Finally, The careful application of selection criteria and assessment procedures has enhanced the internal validity, but additional studies are required to improve the external validity [34].
Implications: Frail older adults seem to have lost the ability to compensate adequately for suboptimal gait parameters, resulting in a reduction of gait speed. We propose examining the compensatory capacity of other biomechanical and physiological regulatory mechanisms to determine whether a transition from compensation to decompensation is a common characteristic of frailty. If so, this could provide an alternative approach to studying frailty in a systemic and dynamic manner.
We have included this information in a new section in the discussion on lines 677-698.

Round 2
Reviewer 2 Report (New Reviewer)
Well done!
Author Response
Manuscript ID: ijerph-2095577 “Frailty syndrome as a transition from compensation to decompensation: Application to the biomechanical regulation of gait”
Response to Reviewer 2
Comment: Well done!
Response: Thank you for your comments, which helped us to improve our manuscript.

Reviewer 3 Report (New Reviewer)
Although the manuscript was modified and some details have improved, unfortunately, a lot of detail for this research is not clearly presented in the manuscript.
Abstract:
Please specific the purpose, methods, results and discussion of this research. A lot of information were miss such participants? group? instrument etc. Please refer to the abstracts of relevant quantitative studies before rewriting.
Introduction
Lines 35-36: What methodologies were used? Is there any literature support for the above?
Lines 36-38: Is there any literature support for the above?
Lines 74-76: Where is the literature?
Lines 81-84: Is there any literature support for the above?
Lines 86-87: Is there any literature support for the above?
Lines 90-91: Where is the literature?
Lines 95-100: I suggest this belong to the method part
Methods
Line 141-144: Is there any literature support for the above?
Line 146-153:“In further studies….” Should in research limitation or conclusions.
Table 1: What's purpose of compared the groups? Here is no any statistical describe.
Line 226: “[44]]” Please revise.
Results
Table 2-4. Please give clear indication of the symbol and also adjust the format and layout of the table appropriately.
Discussion/Conclusions
Line 341-344: Is there any literature support for the above or your inference based on the results?
Line 345-361: The entire content seems to be just a restatement of the results without much specific discussion or comparison. It is recommended that you try to delve deeper into the results you are describing, such as exploring possible causes, impacts, limitations, or potential applications.
Line 362-394: The entire content seems to be like a literature review, however its needs exploring possible causes, impacts, limitations, or potential applications from the results. Please shows the major finding and discuss.
Line 408-410: Where is the literature? Please cite.
Line 410-412: Is there any literature support for the above or your inference based on the results?
Line 423-452: This paragraph seems unnecessary and unrelated to the study. Even if the author intended to explain Compensation and decompensation, it should have been done in the Introduction. This paragraph does not provide any meaningful discussion of the results and findings of the study.
Line 466-486: I suggest some concept of the paragraph could be mention in conclusions.
Author Response
Manuscript ID: ijerph-2095577 “Frailty syndrome as a transition from compensation to
decompensation: Application to the biomechanical regulation of gait”
Response to Reviewer 3
(Please see the attachment)
Abstract:
Please specific the purpose, methods, results and discussion of this research. A lot of information were miss such participants? group? instrument etc. Please refer to the abstracts of relevant quantitative studies before rewriting.
We have corrected the abstract, following the suggestions of the referee. The abstract now follows the structure: introduction, methodology, results, discussion and conclusion, and gives details on the study groups and the measurement device used. The abstract now follows the format of our previously published article in the same journal [Álvarez-Millán, L.; Lerma, C.; Castillo-Castillo, D.; Quispe- Siccha, R.M.; Pérez-Pacheco, A.; Rivera-Sánchez, J.; Fossion, R. Chronotropic Response and Heart Rate Variability before and after a
160 m Walking Test in Young, Middle- Aged, Frail, and Non-Frail Older Adults. Int. J. Environ. Res. Public Health2022,19,8413. https:// doi.org/10.3390/ijerph19148413]
See lines 16-35 in the manuscript:
Abstract: [INTRODUCTION] Most gait parameters decrease with age and are even more importantly reduced with frailty. However other gait parameters exhibit different or even opposite trends for aging and frailty, and the underlying reason is unclear. Literature focuses either on aging, or on frailty, and a comprehensive understanding of how biomechanical gait regulation evolves with aging and with frailty seems to be lacking. [METHODOLOGY] We monitored gait dynamics in young adults (19–29 years, n=27, 59% women), middle-aged adults (30–59 years, n = 16, 62% women), and non-frail (>60 years, n = 15, 33% women) and frail older adults (>60 years, n = 31, 71% women) during a 160m walking test using the triaxial accelerometer of the Zephyr Bioharness 3.0 device. Frailty was evaluated using the Frail Scale (FS) and the Clinical Frailty Scale (CFS). [RESULTS] We found that in non-frail older adults, certain gait parameters, such as cadence, were increased, whereas other parameters, such as step length, were decreased, and gait speed is maintained. Conversely, in frail older adults, all gait parameters, including gait speed, were decreased. [DISCUSSION] Our interpretation is that non-frail older adults compensate for a decreased step length with an increased cadence to maintain a functional gait speed, whereas frail older adults decompensate and consequently walk with a characteristic decreased gait speed. We quantified compensation and decompensation on a continuous scale using ratios of the compensated parameter with respect to the corresponding compensating parameter. [CONCLUSION] Compensation and decompensation are general medical concepts that can be applied and quantified for many, if not all, biomechanical and physiological regulatory mechanisms of the human body. This may allow for a new research strategy to quantify both aging and frailty in a systemic and dynamic way.
Introduction
Lines 35-36: What methodologies were used? Is there any literature support for the above?
The “multiple methodologies” that we mentioned, referred to the 3 existing approaches to frailty. We know cite explicitly Ref. [3] that first defined these 3 different approaches in the manuscript:
(lines 43-45) As discussed in Ref. [3] and explained more into detail here below, frailty has been studied using multiple methodologies, each focusing on different facets of the syndrome. The original research approach to frailty is clinical and empirical, …
(line 55-56) … A second approach to frailty A second approach to frailty takes a basic medical science perspective, …
(lines 61-65) … Therefore, inspired by the field of dynamical systems from physics and mathematics, a third approach incorporates the stimulus-response paradigm…
Lines 36-38: Is there any literature support for the above?
We now give the explicit reference, Ref. [4], see lines 45-47 in the manuscript:
The original research approach to frailty is clinical and empirical, where frailty-related signs and symptoms in the patient are quantified using standardized and validated clinical scales [4].
Lines 74-76: Where is the literature?
The references were in the text below the lines given by the referee. We now state this explicitly in the text, see lines 84-87 in the manuscript:
As explained more into detail here below, much is known in the literature on how gait parameters evolve with aging or frailty, particularly using accelerometry that may be applied in an ambulatory way to reflect activities of daily living (ADL).
Lines 81-84: Is there any literature support for the above?
We now give the explicit Ref. [29], see lines 92-94 in the manuscript:
The walk ratio or step-length-vs.-step-frequency ratio is a speed-independent index of overall neuromotor gait control, reflecting energy expenditure, balance, between-step variability, and attentional demand [29].
Lines 86-87: Is there any literature support for the above?
The references of the lines referred to by the referee are the same as those of the next phrase. This is now clearly stated in the manuscript on lines 96-99:
Acceleration magnitude quantified by the root mean square (RMS) is reduced along the anteroposterior and vertical axes with aging and with frailty [31-33]. The RMS is reduced also along the mediolateral axis with frailty [31] but is reduced less [32] or may even be increased with aging [33].
Lines 90-91: Where is the literature?
The lines 101-102 of the manuscript:
It is clear from the literature that most gait parameters decrease with aging and even more so with frailty.
Refer to the paragraph immediately above, lines 87-99, this should be clear from the context. We did not apply any changes:
Average walking speed gradually decreases with age and is reduced below < 0.8 m/s with frailty [1,24,25]. It is also well documented, although poorly understood, that step and stride length decrease with aging [26] and with frailty [27]. Cadence is reduced with frailty, but the trend is less clear for aging, where some studies suggest that cadence is maintained [25], whereas other studies even find an increased cadence [28]. The walk ratio or step-length-vs.-step-frequency ratio is a speed-independent index of overall neuromotor gait control, reflecting energy expenditure, balance, between-step variability, and attentional demand [29]. The walk ratio is not necessarily related to aging [30], but has been associated with the risk of falling [25] and is not usually applied in frailty research. Acceleration magnitude quantified by the root mean square (RMS) is reduced along the anteroposterior and vertical axes with aging and with frailty [31-33]. The RMS is reduced also along the mediolateral axis with frailty [31] but is reduced less [32] or may even be increased with aging [33].
Lines 95-100: I suggest this belong to the method part
In previous revisions of the manuscript, there were at least 2 referees who insisted explicitly that we should clearly state the research hypothesis, research question and objectives of our study. This is the reason why we included some methodologic details at the end of the introduction section. We left lines 107-110 of the manuscript of the manuscript unchanged:
In this contribution, we report on the results of a gait experiment where multiple study groups, including young and middle-aged control adults and frail and non-frail older adults, walked 160 m at a self-selected speed while being monitored using triaxial accelerometry.
Methods
Line 141-144: Is there any literature support for the above?
We refer now to our previous publication in the same journal, Ref. [38], see lines 157-166 of the manuscript:
The inclusion and exclusion criteria have been explained in a previous publication [38] and are summarized here. To obtain a more realistic sample of frailty within the older adult population, we included individuals of both genders and with diverse medical histories. The exclusion criteria were Parkinson's disease, muscular dystrophy, postural vertigo, as well as physical, auditory, visual, or vestibular limitations that put them at risk during the exercise test. For the same reason, people with heart disease, such as severe cardiac failure, severe supraventricular alteration or monomorphic ventricular extrasystole, were excluded. Exclusion criteria did not include specific alterations such as sarcopenia or cognitive, cardiovascular, or metabolic alterations, instead, we considered such alterations as possible components of the frailty syndrome.
Line 146-153:“In further studies....” Should in research limitation or conclusions.
We removed these lines from the Methods section of the manuscript. As the referee suggested, part of this text is now mentioned in the Limitations subsection , see lines 509-511 of the manuscript:
Limitations: The sample size of our study was relatively small, with only 89 participants. Given that populations of non-frail and frail older adults are rather heterogeneous, a larger population size might be needed to improve the external validity.
Table 1: What's purpose of compared the groups? Here is no any statistical describe.
The referee correctly detected that we did not discuss Table 1 nor its statistical comparisons in the main text. We have now added a new paragraph in the Results section, see lines 286-294 of the manuscript:
Table 1 shows results for demographic and anthropometric variables. By construction, age is significantly different between the four study groups. The control groups of young and middle-aged adults are balanced with respect to male and female participants. In the group of non-frail older adults, the majority of participants are males, whereas in the group of frail older adults the majority are females, which may reflect the fact that frailty has a higher prevalence in women than in men [52]. There are no statistically significant differences between the groups with respect to weight or BMI. Instead, height is significantly decreased in frail older adults (group F) with respect to young adults (group C1), illustrating the importance to consider normalized gait parameters as outlined in subsection 2.4.
Line 226: “[44]]” Please revise.
The typo detected by the referee has now been corrected, see lines 243-245 of the manuscript:
However, other articles show the importance of taking longer walking trials as they capture the slow recovery from perturbations [49].
Results
Table 2-4. Please give clear indication of the symbol and also adjust the format and layout of the table appropriately.
All tables now have a legend explaining the symbols indicating the statistical significance of the pairwise comparisons. The format and the layout of the Tables has been corrected.
The legend added to Tables 2-4 is the following:
¶ p < 0.05 compared to C1 # p < 0.05 compared to C2 § p < 0.05 compared to nF
Discussion/Conclusions
Line 341-344: Is there any literature support for the above or your inference based on the results?
This was a repetition of the lines 92-94 of the manuscript. We have now deleted that specific phrase.
Line 345-361: The entire content seems to be just a restatement of the results without much specific discussion or comparison. It is recommended that you try to delve deeper into the results you are describing, such as exploring possible causes, impacts, limitations, or potential applications.
The purpose of the paragraph mentioned by the referee was to compare the results obtained in our gait experiment with published data by other authors on the same gate parameters. We now state this more explicitly in the manuscript as a “short review” of “what has been reported in the literature” to show that it agrees with our results, see lines 362-364 in the manuscript:
The numerical values for the gait parameters obtained in Tables 2-4 and Figs. 2-4 agree with what has been reported in the literature, as will be illustrated in the short review in the following.
Line 362-394: The entire content seems to be like a literature review, however its needs exploring possible causes, impacts, limitations, or potential applications from the results. Please shows the major finding and discuss.
We disagree, the paragraphs mentioned are not a literature review but discuss the data of Tables 2-4 and Figs. 2-4 of the manuscript and answer the research questions as explicitly stated in the Introduction section. Lines 392-439 of the corrected manuscript discuss why gait parameters have opposite trends for aging and for frailty based on our hypothesis of compensation and decompensation, and lines 440-461 discuss how to quantify compensation and decompensation in contrast to the literature where these phenomena are treated only conceptually or qualitatively.
Line 408-410: Where is the literature? Please cite.
We now give explicitly Ref. [29] on lines 444-446:
The walk ratio l/c has been interpreted as a measure of the quality of overall neuromotor gait regulation [29], and the same interpretation may be valid as well for ΔaAP/ΔaML and ΔaVT/ΔaML given their very similar behaviour.
Line 410-412: Is there any literature support for the above or your inference based on the results?
We rewrote the paragraph and hope our ideas are more clearly stated now, see lines 440-452:
In contrast to the opposite trends of the gait parameters c and ΔaML with aging and frailty discussed so far, the ratios of gait parameters l/c, ΔaAP/ΔaML, and ΔaVT/ΔaML of Table 4 and Fig. 4 show a monotonous decreasing trend over all four populations from no compensation (groups C1 and C2) to compensation (group nF) and finally to decompensation (group F). The walk ratio l/c has been interpreted as a measure of the quality of overall neuromotor gait regulation [29], and the same interpretation may be valid as well for ΔaAP/ΔaML and ΔaVT/ΔaML given their very similar behaviour. Here, we propose the ratios l/c, ΔaAP/ΔaML, and ΔaVT/ΔaML as concrete and practical metrics to quantify compensation and decompensation in gait dynamics. The rationale is the following: each of these ratios evaluates a gait parameter that is being compensated with respect to the gait parameter that is actively performing the compensation, therefore both an increased compensation and an underperforming regulatory mechanism (decompensation) result in a reduced ratio.
Line 423-452: This paragraph seems unnecessary and unrelated to the study. Even if the author intended to explain Compensation and decompensation, it should have been done in the Introduction. This paragraph does not provide any meaningful discussion of the results and findings of the study.
We thank the referee for the suggestion. We moved a shortened version of the paragraph to the introduction, see lines 114-121:
A well-known example from cardiology is compensated heart failure (CHF) after myocardial infarction, where mean systemic blood pressure is increased to compensate for a decreased diastolic filling capacity to ensure a functional cardiac output [35]. Stable chronic heart failure may easily decompensate when specific risk factors place additional strain on the heart muscle and the resulting condition of decompensated heart failure (DHF) implies a worsening of the symptoms including dyspnea, swelling of lower extremities and fatigue [36].
Line 466-486: I suggest some concept of the paragraph could be mention in conclusions.
Given that the section of Strengths, Limitations, and Implications and the section of Conclusions follow each other immediately, we preferred not to repeat the ideas of the former in the latter.

This manuscript is a resubmission of an earlier submission. The following is a list of the peer review reports and author responses from that submission.
Round 1
Reviewer 1 Report
1. The authors are requested to provide a diagram of their test apparatus, with a schematic of their positioning on a sample test subject.
2. The "design and procedure" segment mentions:
"The participants completed the trajectory in one direction, after which they turned around to complete the trajectory again in the other direction, to cover a total distance of L=160m [12]."
How did the authors deal with measurement bias? Generally, it involves taking multiple reading at each test condition, followed by averaging. The text of does not reflect implementing any such measures. Can the authros please elaborate?
3. Were the test subjects made to undergo any king of pre-conditioning cycle prior to the actual experiment?
4. It would be better to move the paragraph starting at line 169 and ending at 182 to the "Design and procedure" section.
Author Response
Comment: The authors are requested to provide a diagram of their test apparatus, with a schematic of their positioning on a sample test subject.
Response: Thank you for your comments, which helped us to improve our manuscript. On page 6/17 (lines 294 to 295), we have included in the manuscript a figure with the requested diagram.
Comment: The "design and procedure" segment mentions:
"The participants completed the trajectory in one direction, after which they turned around to complete the trajectory again in the other direction, to cover a total distance of L=160m [12]."
How did the authors deal with measurement bias? Generally, it involves taking multiple reading at each test condition, followed by averaging. The text of does not reflect implementing any such measures. Can the authros please elaborate?
Response: We did not repeat the gait test for the following reason:
The gait parameters reported in the manuscript are calculated from the triaxial accelerometry time series. The time series reflect the specific characteristics of individual steps and movements for each participant.
We are reporting gait parameters that reflect a time-average of all the observations over the whole time series. If the time series is short, the statistics of the average could be biased, however, for larger time series, the statistics will be more robust.
One of the most popular devices to analyze gait parameters is the GAITRite carpet, which has a length of 9.84m. Gait experiments involving the GAITRite usually involve multiple measurements to establish more precise average values of gait parameters. In our experiment, using the ambulatory Zephyr Bioharness device, we have observations gait dynamics over 160m, consisting of 200 to 500 steps, which allows to establish reliable average values for the gait parameters.
We have included this explanation in the manuscript in lines 309 to 313: “The participants walked the trajectory in one direction, after which they turned around to complete the trajectory again in the other direction, to cover a total distance of L=160m. The experiment was realized once for each participant. Average gait parameters were calculated for the total trajectory that typically includes between 200 to 500 steps.
Comment: Were the test subjects made to undergo any king of pre-conditioning cycle prior to the actual experiment?
Response: The participants did not realize a pre-conditioning cycle for two reasons. First, the test is very simple and reflects activities from daily life such as walking one or more blocks to a local shop and does not require any specialized practice. Secondly, the study population includes frail older adults and any additional activities were avoided that might increase stress or fatigue in this vulnerable population.
We have included this information in lines 313 to 317: “The participants did not realize a pre-conditioning cycle for two reasons. First, the test is very simple and reflects activities from daily life such as walking one or more blocks to a local shop and does not require any specialized practice. Secondly, the study population includes frail older adults and any additional activities were avoided that might increase stress or fatigue in this vulnerable population.”
Comment: It would be better to move the paragraph starting at line 169 and ending at 182 to the "Design and procedure" section.
Response: We agree, and we have moved the section as requested.
See the new subsection “2.1. Description of participant groups”, on lines 254 to 272.

Reviewer 2 Report
Oct. 2, 2022
Alvarez-Millan et al.
Frailty syndrome as a transition from compensation to decompensation: Application to the biomechanical regulation of gait
The study collected and compared biomechanical data on gait among several groups - young and mature control adults, and frail and no frail older adults. The goal of the study was to better understand how dysregulation develops and advances with aging and frailty. However, there are no specific research questions or hypotheses offered. Thus, the results and discussion are impossible to evaluate.
Introduction
L 34: It would be helpful if the authors would provide the definition of frailty they used.
L 35, 39 Authors should define regulation and dysregulation.
L 38 The quote should include p. # in the citation.
Title of paper is “Frailty syndrome as a transition from compensation to decompensation…”, yet background regarding compensation and decompensation is completely absent from the introduction. In fact, decompensation is not mentioned until the final paragraph of the paper.
The authors indicate that “it has been suggested that that frailty should be studied from a complex system approach…”, thereby implying that frailty has not been studied as such previously. The authors should elaborate on what they mean by a complex-system approach, and also convince the reader that such an approach is necessary. Moreover, it is not clear how their “gait experiment” constitutes a complex-system approach. Specific research questions and associated hypotheses are completely absent. In short, the authors fail to provide a compelling rationale for the study.
Methods
L 116: What activity did the participant engage in? - walking, I assume. Should state that the participants walked.
Results
L 163 The entire section 3.1 Description of Participant Groups belongs in the methods section 2.1, not the results. Participant characteristics are not results.
Discussion
L 237 Begins by discussing dysregulation. However, dysregulation was never defined.
L 237-254 Not sure of the purpose of the first paragraph of the discussion, which essential reviews literature on gait parameters. This information belongs in the intro.
Overall, the results and discussion are impossible to evaluate because the authors fail to provide a compelling rationale for conducting the study, and fail to offer any specific research questions or hypotheses. This study appears to have consisted entirely of collecting a bunch of gait data and comparing it among different groups to see what turned up.
Author Response
Comment: The study collected and compared biomechanical data on gait among several groups - young and mature control adults, and frail and no frail older adults. The goal of the study was to better understand how dysregulation develops and advances with aging and frailty. However, there are no specific research questions or hypotheses offered. Thus, the results and discussion are impossible to evaluate.
Response: Thank you for your comments, which helped us to improve our manuscript. We have included the hypothesis and the objective more explicitly in the last 2 paragraphs of the introduction section.
The hypothesis is as follows (lines 75 to 94):
“In the present contribution, we are interested in generic aspects of regulatory mecha-nisms. Our focus will not be the recuperation times of these mechanisms, but the inter-play between multiple variables that play different role. From the point of view of con-trol theory, the essence of homeostatic regulation is a negative feedback loop [13]. The value of a regulated variable, such as blood pressure, is approximately maintained constant near a predefined setpoint. This constancy is the result of the coordinated adaptive responses of multiple effector variables, such as heart rate, ejection fraction and peripheral resistance, to perturbations from the environment [10,14–16]. In early stages of dysregulation, sometimes called unstable homeostasis or homeostatic effort [17], one effector variable may compensate for other underperforming effector variables with as an objective to maintain the associated regulated variable near the optimal val-ue. Compensation requires additional energy which may be unsustainable, and when in the long run the system decompensates, the regulated variable as a consequence deviates from its normal range. These patterns of compensation between effector varia-bles and/or regulated variables that deviate from a normal range, may be universal and applicable to any regulatory mechanism. If true, then evaluating these patterns over the most representative regulatory systems of an older adult, such as the cardio-vascular, respiratory, metabolic and biomechanical systems, may allow to understand not only the heightened vulnerability to stressors for a specific mechanism, but also would allow to integrate over multiple systems and to obtain a whole-body evaluation of the frailty status.”
The study objective is as follows (lines 119 to 122):
“The objective of the present contribution is to investigate the interplay of the before-mentioned gait parameters from the perspective of a regulatory mechanism and to ana-lyze how dysregulation advances with aging and with frailty.”
Introduction
Comment:
L 34: It would be helpful if the authors would provide the definition of frailty they used.
Response: Following your suggestions, we have re-written the introduction. We used the theoretical definition in lines 45-47 and the clinical quantification we used in our study is described in the methodology section on lines 136 to 146. However, for a more complete discussion about frailty syndrome see lines 48 to 74.
Comment:
L 35, 39 Authors should define regulation and dysregulation.
Response: We have included a discussion about regulation and dysregulation in the introduction section on lines 77 to 88.
“From the point of view of control theory, the essence of homeostatic regulation is a neg-ative feedback loop [13]. The value of a regulated variable, such as blood pressure, is approximately maintained constant near a predefined setpoint. This constancy is the re-sult of the coordinated adaptive responses of multiple effector variables, such as heart rate, ejection fraction and peripheral resistance, to perturbations from the environment [10,14–16]. In early stages of dysregulation, sometimes called unstable homeostasis or homeostatic effort [17], one effector variable may compensate for other underperform-ing effector variables with as an objective to maintain the associated regulated variable near the optimal value. Compensation requires additional energy which may be un-sustainable, and when in the long run the system decompensates, the regulated varia-ble as a consequence deviates from its normal range.”
Comment:
L 38 The quote should include p. # in the citation.
Response: Due to the changes in introduction, this quote no longer appears.
Comment:
Title of paper is “Frailty syndrome as a transition from compensation to decompensation…”, yet background regarding compensation and decompensation is completely absent from the introduction. In fact, decompensation is not mentioned until the final paragraph of the paper.
Response: The concepts of compensation and decompensation are now discussed in the introduction section, together with the concepts of regulation and dysregulation, on the same lines 83 to 90 (see here above).
Comment: The authors indicate that “it has been suggested that that frailty should be studied from a complex system approach…”, thereby implying that frailty has not been studied as such previously. The authors should elaborate on what they mean by a complex-system approach, and also convince the reader that such an approach is necessary. Moreover, it is not clear how their “gait experiment” constitutes a complex-system approach. Specific research questions and associated hypotheses are completely absent. In short, the authors fail to provide a compelling rationale for the study.
Response: In the original manuscript, the concept of complex systems was only mentioned twice. In this revised version of the manuscript, we have removed the concept of complex systems.
Methods
Comment:
L 116: What activity did the participant engage in? - walking, I assume. Should state that the participants walked.
Response: We have changed the word completed for walking to be more specific.
(Line 309)
“The participants walked the trajectory in one direction, after which they turned around to complete the trajectory again in the other direction, to cover a total distance of L=160m [12].”
Results
Comment:
L 163 The entire section 3.1 Description of Participant Groups belongs in the methods section 2.1, not the results. Participant characteristics are not results.
Response: We have moved the section 3.1 to methods section 2.1(Lines 254 to 272)
Discussion
Comment:
L 237 Begins by discussing dysregulation. However, dysregulation was never defined.
Response: We have included a discussion about regulation and dysregulation in the introduction section on lines 77 to 88 (see here above).
Comment:
L 237-254 Not sure of the purpose of the first paragraph of the discussion, which essential reviews literature on gait parameters. This information belongs in the intro.
Response: We have moved the first paragraph of the discussion section of the original manuscript to the last paragraph of the introduction section of the revised manuscript. See lines 106 to 119.
“Much is known in the literature on accelerometry-derived gait parameters. Average walking speed gradually decreases with age and is reduced below the functional level <0.8 m/s with frailty [8,25,26]. It is also well documented, although poorly understood, that step and stride length decrease with aging [27] and with frailty [28]. Cadence is reduced with frailty, but the trend is less clear for aging, where some studies suggest that cadence is maintained [25], whereas other studies even find an increased cadence [29]. The walk ratio, or step-length-vs.-step-frequency or cadence ratio, is a speed-independent index of the overall neuromotor gait control, reflecting energy ex-penditure, balance, between-step variability, and attentional demand. The walk ratio is not necessarily related to aging [22] but has been associated to the risk of falling [25] and is not usually applied in frailty research. Acceleration magnitude as quantified by the root mean square (rms) is reduced along the anteroposterior and the vertical axes with aging and with frailty. The rms may be increased along the mediolateral axis with aging [30,31] but not with frailty [32].”
Comment:
Overall, the results and discussion are impossible to evaluate because the authors fail to provide a compelling rationale for conducting the study, and fail to offer any specific research questions or hypotheses. This study appears to have consisted entirely of collecting a bunch of gait data and comparing it among different groups to see what turned up.
Response: We have included the hypothesis and the objective more explicitly in the last 2 paragraphs of the introduction section (see here above).

Reviewer 3 Report
The Authors aim to better understand how dysregulation develops in frailty subjects investigating how regulatory mechanisms interplay with various gait parameters, thus providing a possible diagnostic factor for frailty. The study is original and conclusions interesting.
Nevertheless, there are some points to address:
- in the introduction and in the discussion sections more background is needed on frailty, with particular regard to diagnosis, and differences between aging, frailty and sarcopenia.
- since aging, sarcopenia and frailty can overlap, can the authors exclude that in old group (nF and F) some sarcopenic subjects is present? Did the Authors investigate for that?
- why the authors divided the old subjects in only two groups? I would add and additional group of subjects having >70yrs, to see if the investigated parameters decrease further after that age.
- the authors did not investigate about possible muscle impairment in the enrolled population. Only muscle dystrophy has been considered in the exclusion criteria. Since gait is influenced by muscle mass and function, muscle condition in terms of atrophy (for example by CT scan or DEXA), force and overt myopathies should be addressed. Information about drugs or therapies (i.e. statins, corticosteroids, or chemotherapy) known to induce myopathies should also be integrated (including their cumulative dosage).
- did the authors have information about weight loss? If yes, please include this data in Table 1 and also as additional variable in the statistical analyses.
- in Figures 1 and 2 please include also asterisks for statistical significance.
Author Response
Comment: in the introduction and in the discussion sections more background is needed on frailty, with particular regard to diagnosis, and differences between aging, frailty and sarcopenia.
Response: Thank you for your comments, which helped us to improve our manuscript. Following your suggestions, we have re-written the introduction with a more extensive discussion on aging and frailty (lines 34 to 74) and we now state explicitly in the materials and methods section that sarcopenia was not part of the exclusion criteria (lines 150 to 153).
Comment: since aging, sarcopenia and frailty can overlap, can the authors exclude that in old group (nF and F) some sarcopenic subjects is present? Did the Authors investigate for that?
Response: One of the common aspects between aging, frailty and sarcopenia are gait alterations. Being the gait speed one of the affected parameters. However, there are many different types of alterations that can lead to the slowing down of gait, for instance, physical, cognitive, sensorial, cardiovascular or metabolic alterations (Fritz, S., & Lusardi, M. (2009)).
On the other hand, frailty is a multisystemic syndrome in which several systems are altered (Clegg, A., & Young, J. (2011)). Therefore, to exclude possible causes that lead to the slowing down could include sarcopenia but also the other alterations mentioned. However, excluding that many cases would lead to a misrepresented study as is the set of all these alterations that leads older adults to be frail.
Interestingly, frailty syndrome has shown that regardless of the patient's condition(s), all the frail older adults walk slowly.
We added the following text to the manuscript (lines 150 to 156):
“We did not consider as exclusion criteria or as different subgroups some of the specific alterations that can lead to a slower gait, e.g., sarcopenia or cognitive, cardiovascular or metabolic alterations, etc., and considered these as possible components of the frailty syndrome. Independent from the original causes of frailty in an individual, one of the generic effects of frailty is a slower walking speed, and we were interested to see whether we could detect similar generic alterations at the level of gait regulation.”
Comment: why the authors divided the old subjects in only two groups? I would add and additional group of subjects having >70yrs, to see if the investigated parameters decrease further after that age.
Response: It would be very interesting to study more specific populations in order to analyze the effect of loss of muscle strength (for instance). Unfortunately, the sample size of this study is not enough to divide the groups, as the resulting subgroups would not be representative.
We included the following text in lines 156 to 252:
“In further studies, it would be desirable to increase the population size to analyze the gait parameters for different subgroups, for instance, to study a possible effect of loss of muscle strength in older adults over 70 years old.”
Comment: the authors did not investigate about possible muscle impairment in the enrolled population. Only muscle dystrophy has been considered in the exclusion criteria. Since gait is influenced by muscle mass and function, muscle condition in terms of atrophy (for example by CT scan or DEXA), force and overt myopathies should be addressed. Information about drugs or therapies (i.e. statins, corticosteroids, or chemotherapy) known to induce myopathies should also be integrated (including their cumulative dosage).
Response: The focus of this article is not to study the causes of gait alterations, whether loss of muscle mass or function, alterations in equilibrium or eyesight, etc.; we are interested in the effects of frailty on gait regulation independent from the original causes of frailty. It is well known that all frail older adults walk slowly. Although, in a study with a higher number of participants, it would be very interesting to study particular subgroups.
The text included for this comment is the same of previous comment (in lines 156 to 252).
Comment: did the authors have information about weight loss? If yes, please include this data in Table 1 and also as additional variable in the statistical analyses.
Response: Unfortunately, we do not have that information.
Comment: in Figures 1 and 2 please include also asterisks for statistical significance.
Response: We have included asterisks in figures 2 and 3.
See Lines 444 to 445 and 465 to 466.
Figure 2. Velocity (v), normalized step length (ln) and normalized cadence (cn) for young adults (C1), mature adults (C2), non-frail older adults (nF) and frail older adults (F) calculated along the time axis (a-c) and total (Δa), anteroposterior (ΔaAP), vertical (ΔaVT) and mediolateral (ΔaML) step acceleration, along the amplitude axis (d-f). (a-f) show mean ± SE for each value. The pairwise statistically significant differences are indicated in the figure. p<0.05 (*), p<0.005 (**), p<0.001 (***). Horizontal gridline indicate the gait velocity threshold (v=0.83 m/s) obtained with a ROC curve for the frail older adults (F) of this study.
Figure 3. Ratios of gait parameters calculated along the time axis (a) ln/cn and along the amplitude axis (b) ΔaAP/ΔaML and (c) ΔaVT/ΔaML for young adults (C1), mature adults (C2), non-frail older adults (nF) and frail older adults (F). The pairwise statistically significant differences of Table 2 are indicated in the figure. p<0.05 (*), p<0.005 (**), p<0.001 (***).

Round 2
Reviewer 2 Report
Unfortunately, as in the original submission, specific research questions and associated hypotheses are absent. The authors only generally characterize the objective of the study as follows:
“The objective of the present contribution is to investigate the interplay of the before-mentioned gait parameters from the perspective of a regulatory mechanism and to ana-lyze how dysregulation advances with aging and with frailty.”
This is insufficient. An introduction must offer specific research questions (in the form of questions) and associated hypotheses. The authors claim to have included
the hypothesis and the objective more explicitly in the last 2 paragraphs of the introduction section.
I repeatedly read these paragraphs and could not find any hypotheses offered for specific research questions.
Reviewer 3 Report
I'm satisfied by the revision submitted.